# A novel customer churn prediction model for the telecommunication industry using data transformation methods and feature selection

Joydeb Kumar Sana[1], Mohammad Zoynul Abedin[2], M. Sohel Rahman[1], M. Saifur Rahman[1] *

**1** Department of Computer Science and Engineering, Bangladesh University of Engineering and Technology, Dhaka, Bangladesh, **2** Department of Finance and Banking, Hajee Mohammad Danesh Science and Technology University, Dinajpur, Bangladesh

* mrahman@cse.buet.ac.bd

## Abstract

Customer churn is one of the most critical issues faced by the telecommunication industry (TCI). Researchers and analysts leverage customer relationship management (CRM) data through the use of various machine learning models and data transformation methods to identify the customers who are likely to churn. While several studies have been conducted in the customer churn prediction (CCP) context in TCI, a review of performance of the various models stemming from these studies show a clear room for improvement. Therefore, to improve the accuracy of customer churn prediction in the telecommunication industry, we have investigated several machine learning models, as well as, data transformation methods. To optimize the prediction models, feature selection has been performed using univariate technique and the best hyperparameters have been selected using the grid search method. Subsequently, experiments have been conducted on several publicly available TCI datasets to assess the performance of our models in terms of the widely used evaluation metrics, such as AUC, precision, recall, and F-measure. Through a rigorous experimental study, we have demonstrated the benefit of applying data transformation methods as well as feature selection while training an optimized CCP model. Our proposed technique improved the prediction performance by up to 26.2% and 17% in terms of AUC and F-measure, respectively.

## 1 Introduction

Over the last few decades, the telecommunication industry (TCI) has witnessed enormous growth and development in terms of technology, level of competition, number of operators, new products, services and so on. However, because of extensive competition, saturated markets, dynamic environment, and attractive and lucrative offers, the TCI faces serious customer churn issues, which is considered to be a formidable problem in this regard [1]. In a

**Data Availability Statement:** https://www.kaggle.com/abhinav89/telecom-customer/data (Last Access: September 03, 2022). https://data.world/

**Funding:** The author(s) received no specific funding for this work.

**Competing interests:** The authors have declared that no competing interests exist.

competitive market, where customers have numerous choices of service providers, they can easily switch services and even service providers. Such customers are referred to as churned customers [1] with respect to the original service provider.

The two main generic strategies to generate more revenues in an industry are (i) increase the retention period of customers and (ii) acquire new customers [2]. In fact, customer retention is believed to be the most profitable strategy, as customer turnover severely hits the company's income and its marketing expenses [3].

Churn is an inevitable result of a customer's long term dissatisfaction over the company's services. Complete withdrawal from a service (provider) on part of a customer does not happen in a day; rather the dissatisfaction of the customer, grown over time and exacerbated by the lack of attention by the service provider, results in such a fiery gesture by the customer. To prevent this, the service provider must work on limitations (perceived by the customers) in its services to retain the aggrieved customers. Thus it is highly beneficial for a service provider to be able to identify whether a customer is likely to churn. In this context, non-churn customers are those who are reluctant to move from one service provider to another in contrast to churn customers.

If a telephone company (TELCO) can identify the customers who are likely to churn, then it can potentially cater targeted offerings to them to reduce their dissatisfaction, increase engagement and thus potentially retain them. This has a clear positive impact on revenue. Additionally, customer churn adversely affects the company's fame and branding. As such, churn prediction is a very important task particularly in the telecom sector. To this end, TELCOs generally maintain a detailed standing report of the customers to understand their standing and to anticipate their longevity in continuing the services. Since the expense of getting new customers is relatively high [4], TELCO nowadays principally focuses on retaining their long-term customers rather than getting new ones. This makes churn prediction essential in the telecom sector [5]. With the above backdrop, in this paper, the customer churn prediction (CCP) problem has been revisited as a binary classification problem in which all of the customers are partitioned into two classes, namely, Churn and Non-Churn.

## 1.1 Brief literature review

The problem of CCP has been tackled using various approaches including machine learning models, data mining methods, and hybrid techniques. Several Machine Learning (ML) and data mining approaches (e.g., Rough set theory [3, 6], Naïve Bayes and Bayesian network [7], Decision tree [8, 9], Logistic regression [9], RotBoost [10], Support Vector Machine (SVM) [11], Genetic algorithm based neural network [12], AdaBoost Ensemble learning technique [13], etc.) have been proposed for churn prediction in the TCI using customer relationship management (CRM) data. Notably, CRM data is widely used in prediction and classification problems [14]. A detailed literature review considering all these works is beyond the scope of this paper; however, we briefly review some of the most relevant papers below.

Brandusoiu et al. [15] presented a data mining based approach for prepaid customer churn prediction. To reduce data dimension, the authors applied Principal Component Analysis (PCA). Three machine learning classifiers were used here, namely, Neural Networks (NN), Support Vector Machine (SVM), and Bayes Networks (BN) to predict churn customers. He et al. [16] proposed a model based on Neural Networks (NN) in order to tackle the CCP problem in a large Chinese TELCO that had about 5.23 million customers. Idris et al. [17] proposed a technique combining genetic programming with AdaBoost to model the churn problem in the TCI. Huang et al. [18] studied the problem of CCP in the big data platform. The aim of the

study was to show that big data significantly improves the performance of churn prediction using Random Forest classifier.

Makhtar et al. [19] proposed a rough set theory based model for churn prediction in TCI. Amin et al. [20] on the other hand focused on tackling the data imbalance issue in the context of CCP in TCI and compared six unique sampling strategies for oversampling. Burez et al. [21] also studied the issue of unbalanced datasets in churn prediction models and conducted a comparative study for different methods for tackling the data imbalance issue. Hybrid strategies have also been used for processing massive amount of customer information together with regression techniques that provide effective churn prediction results [22]. On the other hand, Etaiwi et al. [23] showed that their Naïve Bayes model was able to beat a Support Vector Machine (SVM) model in terms of precision, recall, and F-measure.

An important limitation in this context is that most of the methods in the literature have been experimented with on a single dataset. Also, the impact of data transformation methods combined with feature selection on various machine learning classifiers for CCP have not been investigated deeply. There are various DT methods like the Log, Rank, Z-score, Discretization, Min-max, Box-cox, Aarcsine and so on. Among these, researchers broadly used the Log, Z-score, and Rank DT methods in different domains (e.g., software metrics normality and maintainability [24], defect prediction [25], dimensionality reduction [25] etc.). To the best of our knowledge, there are only three works [26–28] in the literature where DT methods have been applied in the context of CCP in TCI. In [26], only two DT methods (Log and Rank) and a single classifier (Naïve Bayes) have been investigated. In [28], two DT methods, Discretization and Weight-of-evidence, have been implemented. However, the authors experimented with only one dataset.

The study by Amin et al. [27] is more recent and is very relevant to our work. Like our study, they too investigate the importance of data transformation methods in the context of the CCP in TCI problem. However, they have used only 1 dataset for model training and another dataset for independent testing. The training dataset has 18000 records. While there are 250 features in this dataset, the dataset for independent testing had 20 features only. Therefore, the overall study was conducted in the context of much smaller sample size and limited set of features. On the other hand, the experiments of this study have been conducted on 4 different datasets. Our results are mostly consistent across the datasets. Dataset-1 in our study has 100000 samples and 101 features. Since the performance of machine learning algorithms generally improves with the amount of data, it is reasonable to expect that our study would produce better models. Indeed this becomes evident when we analyze the results of our experiments and make a comparative analysis with prior works. Besides, the study in [27] did not consider the WOE DT method, which has been shown to be the best performing DT method in the majority of the cases in our experiments. We have also experimented with a more comprehensive list of baseline classifiers compared to their study. Therefore, there is a large room for exploration of DT methods in conjunction with feature selection to come up with optimized machine learning models for CCP in TCI context. This research gap is clearly evident in Table 1. Therefore, in this manuscript, we have endeavored to close this research gap and propose optimized machine learning models which can potentially outperform the state-of-the-art models.

## 1.2 Our contributions

This paper makes the following key contributions:

- Several customer churn prediction models have been developed that leverage various machine learning algorithms and data transformation (DT) methods. In particular, we

**Table 1. Data transformation methods, hyperparameter optimization and feature selection used in prior studies.**

| Study | Data Transformation method | | | | | | Optimization | Feature selection |
|---|---|---|---|---|---|---|---|---|
| | **Log** | **Rank** | **Box-Cox** | **Z-score** | **Discretization** | **WOE** | | |
| Amin et al. [26] | ✓ | ✓ | X | X | X | X | X | X |
| Coussement et al. [28] | X | X | X | X | ✓ | ✓ | X | X |
| Amin et al. [27] | ✓ | ✓ | ✓ | ✓ | X | X | X | ✓ |
| Makhtar et al. [19] | X | X | X | X | X | X | X | X |
| Amin et al. [20] | X | X | X | X | X | X | X | X |
| Burez et al. [21] | X | X | X | X | X | X | X | X |
| Qureshi et al. [22] | X | X | X | X | X | X | X | X |
| Etaiwi et al. [23] | X | X | X | X | X | X | X | X |
| Melian et al. [29] | X | X | X | X | X | X | X | X |
| Andreea et al. [30] | X | X | X | X | X | X | X | ✓ |

✓ (X) mark indicates that the mentioned approach was (was not) used in the study.

have used eight different classifiers combined with six different DT methods to develop a number of models to handle the CCP problem. The classification algorithms used include K-Nearest Neighbor (KNN), Naïve Bayes (NB), Logistic Regression (LR), Random forest (RF), Decision tree (DTree), Gradient boosting (GB), Feed-Forward Neural Networks (FNN), and Recurrent Neural Networks (RNN). On the other hand, the DT methods that have been applied are: Log, Rank, Box-cox, Z-score, Discretization, and Weight-of-evidence (WOE).

- To optimize the machine learning classifiers, univariate technique has been performed to select the most effective features and grid search method has been used to find the best hyperparameters.

- Extensive experiments have been conducted on four different publicly available datasets and our models have been evaluated using various information retrieval metrics such as AUC, Precision, Recall, and F-measure. Our models achieved promising results and our experimental results clearly demonstrate that the DT methods have a positive impact on CCP models.

- Statistical significance tests have also been conducted on our findings. Our results clearly indicate that the impact of DT methods on the classifiers is not only positive but also statistically significant.

- We have provided a concrete decision on the best combination of DT method and prediction model for the CCP in TCI problem. Specifically, we recommend Weight-of-evidence (WOE) for data transformation, followed by model training with Logistic Regression (LR) or Feed-Forward Neural Networks (FNN) to construct a successful CCP model.

## 1.3 Organization of this paper

The rest of the paper is organized as follows. Section 2 illustrates the methodology and the framework of the proposed study. The experimental results are briefly explained in Section 3. Section 4 covers the performance comparison with other studies. The impact of the DT methods on the data distribution is described in Section 5. The discussion section is presented in the Section 6. Finally, we provide concluding remarks in Section 7.

**Table 2. Summary of the datasets used in this study.**

| Description | Dataset-1 | Dataset-2 | Dataset-3 | Dataset-4 |
|---|---|---|---|---|
| No. of samples | 100000 | 5000 | 3333 | 7043 |
| No. of attributes | 101 | 20 | 21 | 21 |
| No. of class labels | 2 | 2 | 2 | 2 |
| % of churn samples | 50.43 | 85.86 | 85.5 | 26.54 |
| % of non-churn samples | 49.56 | 14.14 | 14.5 | 73.46 |
| Data Source | URL[1] | URL[2] | URL[3] | URL[4] |

URL[1]: https://www.kaggle.com/abhinav89/telecom-customer/data (Last Access: September 03, 2022).

URL[2]: https://data.world/earino/churn (Last Access: September 03, 2022).

URL[3]: https://www.kaggle.com/becksddf/churn-in-telecoms-dataset/data (Last Access: September 03, 2022).

URL[4]:https://www.kaggle.com/blastchar/telco-customer-churn (Last Access: September 03, 2022).

# 2 Materials and methods

This section provides a detailed description of the datasets and data transformation methods used in this study. The feature selection technique, hyperparameter tuning, model training, evaluation measures etc. are also discussed in this section.

## 2.1 Datasets

In this study, four publicly available benchmark datasets have been used, that are broadly used for the CCP problem in the telecommunication area. Table 2 briefly describes these datasets.

**2.1.1 Data preprocessing.** The following essential data preprocessing steps have been applied:

- The sample IDs and/or descriptive texts which are used only for informational purposes are ignored.

- Redundant attributes have been removed.

- Missing numerical values were replaced with zero (0) and missing categorical values were treated as a separate category.

- Following previous literature [6], categorical values were encoded such as 'yes' or 'true' as 1, while 'no' or 'false' as 0. For other categorical values, *Label Encoder* (from the *sklearn* python library) was used to generate numeric representation.

## 2.2 Data transformation methods

Data transformation refers to the application of a deterministic mathematical function to each point in a dataset. Table 3 provides a description of the Data Transformation (DT) methods leveraged in our research.

## 2.3 Evaluation measures

The confusion matrix is generally used to assess the overall performance of a predictive model. For the CCP problem, the individual components of confusion matrix is defined as follows: (i) True Positives (TP): correctly predicted churn customers (ii) True Negatives (TN): correctly predicted non-churn customers (iii) False Positives (FP): non-churn customers, miss-predicted as churn customers and (iv) False Negatives (FN): churn customers, miss-predicted as

**Table 3. List of data transformation methods.**

| DT Method | Description | Equation |
|---|---|---|
| Log | Each variable $x$ is replaced with $log(x)$, where the base of the log is left up to the analyst [24]. However, since logarithm of 0 is undefined, therefore in case the feature value contains 0, we have defined the Log transformation to be 0 as well in this study. | $$\text{Log-DT}(x) = \begin{cases} 0 & \text{if } x = 0 \\ ln(x) & \text{if } x > 0 \end{cases} \qquad (1)$$ where $x$ is the value of any feature variable of the original dataset. |
| Rank | It is a statistically calculated rank value [24]. In this research, we have followed the study in [24] to transform the initial values of every feature in the original dataset into ten (10) ranks, using each 10th % (percentile) of the given feature's values. | $$Rank(x) = \begin{cases} 1 & \text{if } x \leq Q_1 \\ k & \text{if } x \in (Q_{(k-1)}, Q_k], k \in \{2, 3.....9\} \\ 10 & \text{if } x > Q_9 \end{cases} \qquad (2)$$ where $Q_k$ is the $(k \times 10)^{th}$ percentile of the corresponding metric. |
| Box-Cox | It is a lambda based power transformation method [24]. This transformation method is a process to transform non-normal feature values into a normal distribution. | $$\text{Box-Cox}(x, \lambda) = \frac{x^2 - 1}{\lambda} \quad \lambda \neq 0 \qquad (3)$$ Where $\lambda$ can be configured by the analyst in the range from -5 to +5, and $x$ is the given value of any feature of the initial dataset. In this study, we have set $\lambda = 0.5$. |
| Z-score | It indicates the distance of a data point from the mean in units of standard deviation [31]. | $$\text{Z-Score} = \frac{x - \text{sample mean}}{\text{sample standard deviation}} \qquad (4)$$ where $x$ is the given value of any feature of the original dataset. |
| Discretization | It is a binning technique [32]. For continuous variables, four widely used discretization techniques are K-means, equal width, equal frequency, and decision tree based discretization. We used the equal width discretization technique which is a very simple method. | For any given continuous variable $x$, the following process is applied: Provided $x_{min}$ is the minimum value of a selected feature and $x_{max}$ the maximum, bin width $\Omega$ can be computed as $$\Omega = \frac{x_{max} - x_{min}}{b} \qquad (5)$$ Hence, the discretization technique generates $b$ bins with boundaries at $x_{min}$, $x_{min} + i \times \Omega$ and $x_{max}$, where $i = 1, 2, \ldots, (b-1)$. $b$ is a parameter chosen by the analyst. |
| Weight-of-evidence (WOE) | It is a binning and log based transformation [28]. In most cases, WOE solves the skewness in the data distribution. | $$WOE_i = ln\left(\frac{X_i}{Y_i}\right) \qquad (6)$$ Where $X_i$ = Proportion of events with bin level $i$ and $Y_i$ = Proportion of non-events with bin level $i$. $ln$ represents natural logarithm. |

non-churn customers. The following popular evaluation measures are used for comparing the performance of the models.

**Precision:** Mathematically, precision can be expressed as:

$$Precision = \frac{TP}{TP + FP} \qquad (7)$$

**The probability of detection (POD)/ Recall:** POD or recall is a valid choice of evaluation metric when we want to capture as many true churn customers as possible. Mathematically POD can be expressed as:

$$Recall/POD = \frac{TP}{FN + TP} \qquad (8)$$

**The probability of false alarm (POF):** The value of POF should be small as much as possible (in an ideal case, the value of POF is 0). Mathematically POF can be defined as:

$$POF/\text{False positive rate} = \frac{FP}{TN + FP} \qquad (9)$$

We use POF for measuring incorrect churn predictions.

**Table 4. List of baseline classifiers.**

| Key | Classifer | Model type | Description |
|-----|-----------|------------|-------------|
| KNN | K-Nearest Neighbor | Instance-based learning, lazy learning | The KNN algorithm assumes that similar things exist in close proximity. |
| NB | Naïve Bayes | Gaussian | NB is a family of probabilistic algorithms. It gives the conditional probability, based on the Bayes theorem. |
| LR | Logistic Regression | Statistical model | Logistic regression is estimating the parameters of a logistic model (a form of binary regression). |
| RF | Random forest | Trees | RF is an ensemble tree-based learning algorithm. |
| DTree | Decision tree | Trees | DTree builds classification or regression models in the form of tree structure. |
| GB | Gradient boosting | Trees | GB is an ensemble tree-based boosting method. |
| FNN | Feed-Forward Networks | Deep learning | FNN is a deep learning classifier where the input travels in one direction. |
| RNN | Recurrent Neural Networks | Deep learning | RNN is a deep learning classifier where the output from previous step are fed as input to the current step. |

**The Area under the Curve (AUC):** Both POF and POD are used to measure the AUC [24, 27]. A higher AUC value indicates a higher performance of the model. Mathematically AUC can be expressed as:

$$AUC = \frac{1 + POD - POF}{2} \tag{10}$$

**F-Measure:** The F-measure is the harmonic mean of the precision and recall. F-measure is needed when we want to seek a balance between precision and recall. A perfect model has an F-measure of 1. The Mathematical formula of F-measure is defined below.

$$\text{F-Measure} = \frac{(2 * precision * recall)}{(precision + recall)} \tag{11}$$

## 2.4 Optimized customer churn prediction models

The baseline classifiers used in our research are presented in Table 4. To examine the effect of the DT methods, we apply them on the original datasets and subsequently, on the transformed data. We have compared the performance in both settings and objectively assessed whether data transformation methods have a positive impact in customer churn prediction in the tele-communication industry.

## 2.5 Validation method and steps

In all our experiments, the classifiers of the CCP models were trained and tested using 10-fold cross-validation on the four different datasets described in Table 2. Firstly, a RAW data based CCP model was constructed without leveraging any of the DT methods on any features of the original datasets. In this case, we did not apply any feature selection steps either. However, we used the best hyperparameters for the classifiers. Subsequently, we applied a DT method on each attribute of the dataset and retrained our models based on this transformed dataset. We experimented with each of the DT methods mentioned in Table 3. For each DT based model, we also used a feature selection and optimization procedure, which is described in the following section.

## 2.6 Feature selection and optimization

Feature selection and hyperparameter optimization techniques have been used in many researches in different fields. However, these issues have not been discussed in the literature of

**Table 5. Hyperparameter optimization results for different classifiers.** All the classifiers are run from the *sklern* Python package.

| Classifier | Optimized hyperparameter values |
|---|---|
| NB | var_smoothing = 1.0 |
| LR | C = 5, multi_class='ovr', penalty='11', solver='liblinear' |
| KNN | metric='manhattan', n_neighbors = 19, weights='uniform' |
| RF | bootstrap = True, max_depth = 80,max_features = 3, min_samples_leaf = 3, min_samples_split = 12, n_estimators = 1000 |
| GB | max_features='sqrt', criterion='mae' |
| DT | criterion='entropy', splitter='best' |
| FNN | batch_size = 10, epochs = 10 |
| RNN | batch_size = 80, epochs = 100 |

CCP in TCI. Feature Selection (FS) [33] is a data pre-processing technique used to find the optimal subset of features that can capture the intrinsic properties of a dataset for the purpose of increasing the learning accuracy. This pre-processing step naturally reduces the dimensionality of the data and allows learning algorithms to operate faster and more effectively. In this study, we used univariate feature selection technique as it is one of the most powerful feature selection techniques, yet easy to compute and simple to interpret [34]. The advantages of the univariate method are that it is fast, scalable and independent of any learning algorithm [33]. Because of this independence from the learning algorithms, we were able to perform the feature selection once and then use the selected features with different prediction algorithms.

Hyperparameter optimization is a systematic process that helps in finding the right hyperparameter values for a machine learning algorithm. In this work, Grid Search (GS) [35] has been used to optimize the parameters of eight classifiers. GS has been widely used in many research works to improve the classification performance, like wind speed forecasting [36], HIV prediction [37], electricity generation prediction [38], cancer cell prediction [39] and so on. The major downside of GS is its ineffectiveness in the configuration space of high dimensional hyperparameters. This is because the number of assessments increases exponentially as the number of hyperparameters increases [37]. However, because of its ease of execution, parallelization and durability in low-dimensional spaces (1-D, 2-D), GS prevails as the state-of-the-art for hyperparameter optimization. For the classifiers used in this study, the hyperparameter space is very low dimensional. As such we have used GS to optimize the the hyperparameter of all the classifiers in this study. Specifically, we have used the *GridSearchCV* method from the *sklearn* python library [35]. Table 5 shows the summary of the best hyperparameter settings for each classifier, while Table 6 represents the time taken to train the classifiers with

**Table 6. Learning time of different classifiers, with hyperparameter optimization using grid search.** The results are shown for Dataset-1, with weight of evidence used as the data transformation method.

| Classifer | Learning Time (seconds) |
|---|---|
| NB | 51 |
| LR | 214 |
| KNN | 1046 |
| RF | 1347 |
| Dtree | 16 |
| GB | 13332 |
| FNN | 1207 |
| RNN | 39305 |

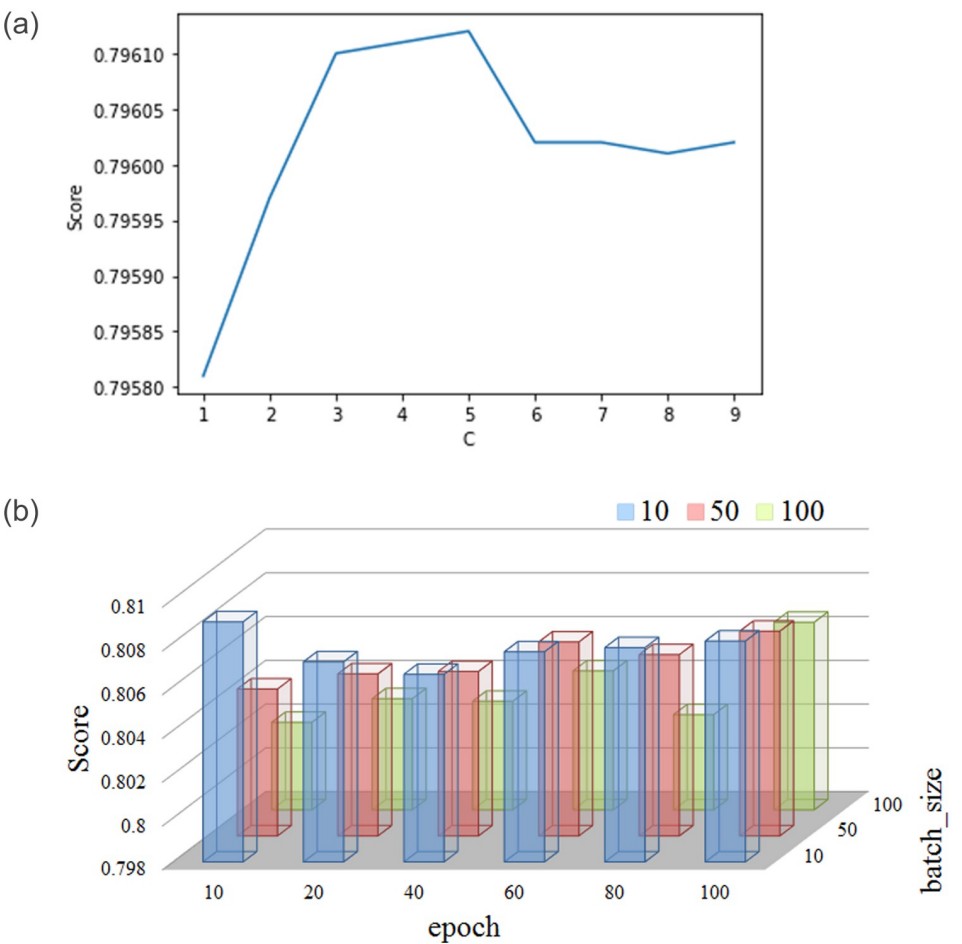

**Fig 1. Hyperparameter optimization using Grid Search. (Left.)** For logistic regression, the parameter C is tuned. The best performance is attained when $C = 5$. **(Right.)** The hyperparameters *epoch* and *batch_size* are optimized for the Feed-forward Neural Network. The best performance is attained when *epoch* = 10, *batch_size* = 10.

hyperparameters optimized using grid search. The behaviour of grid search with respect to the classifiers LR and FNN are depicted in Fig 1.

Fig 2 illustrates the overall flowchart of our proposed optimized CCP model. First, we applied some necessary pre-processing steps on the datasets. Then, DT methods (Log, Rank, Box-cox, Z-score, Discretization, and WOE) were applied. Next, the univariate feature selection technique [40] was used to select the higher scored features from the dataset (we selected the top 80 features for Dataset-1 and top 15 features for the Dataset-2, Dataset-3 and Dataset-4). We applied grid search to find the best hyperparameters for individual classifier algorithms. Finally, 10-fold cross validation was employed to train and validate the models.

## 2.7 Stability measurement tests

We used Friedman non-parametric statistical test (FMT) [41] to examine the reliability of the findings and whether the improvement achieved by the DT based optimized classification models are statistically significant. The Friedman test is the non-parametric statistical test for analyzing and finding differences in treatments across multiple attempts [41]. It does not assume any particular distribution of the data. Friedman test ranks all the methods. It ranks

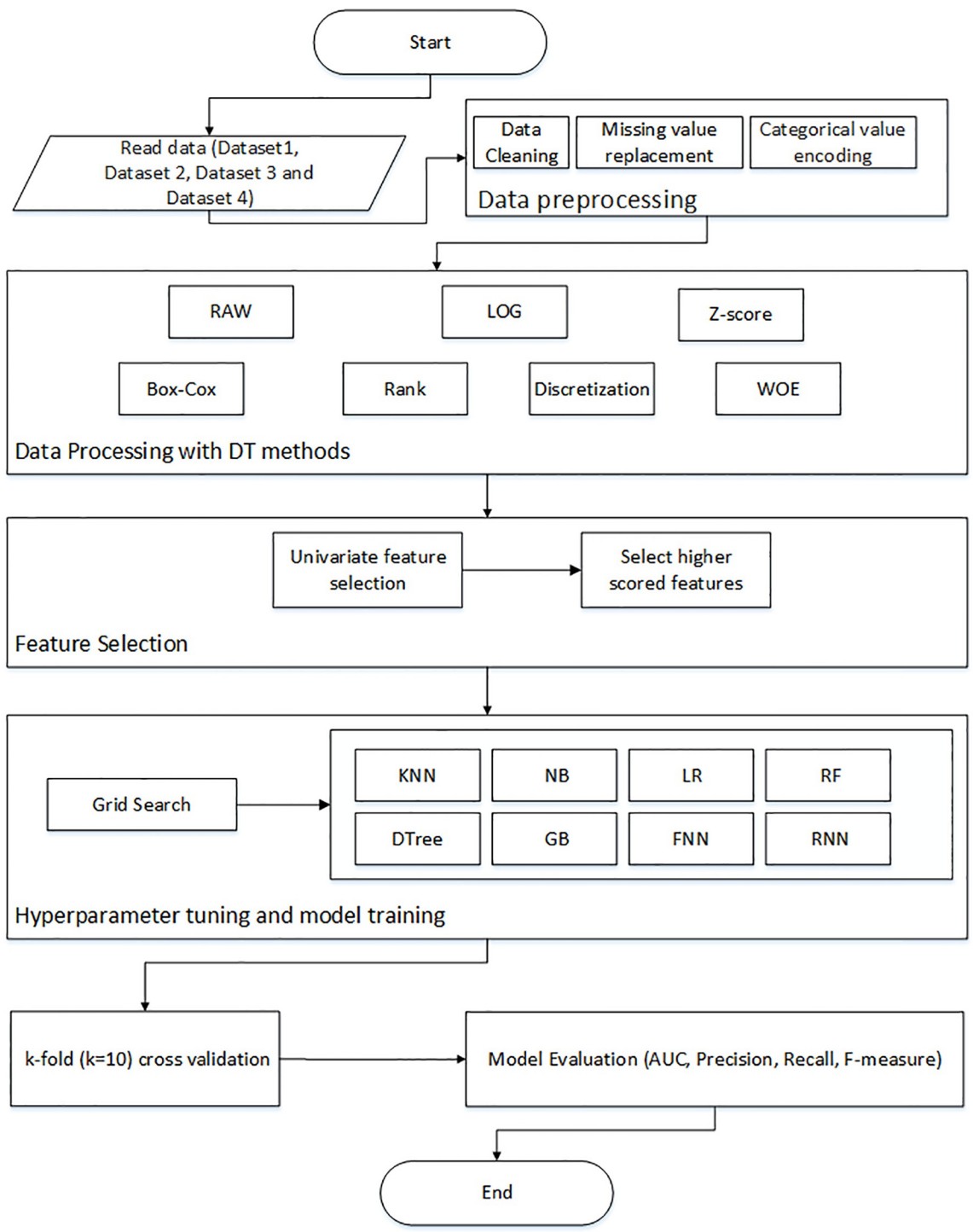

**Fig 2. Flowchart of the optimized CCP model using data transformation methods.**

the classifiers independently for each dataset. Lower rank indicates a better performer. The Friedman test was performed on the F-measure results. Here, the null hypothesis ($H_0$) represents: "there is no difference among the performances of the CCP models". In our experiments, the test was carried out with the significance level, $\alpha = 0.05$.

Subsequently, post hoc Holm test is conducted to perform the paired comparisons with respect to the best performing DT model. In particular, when the null hypothesis is rejected, the post hoc Holm test was used to compare the performance of the models. This test is a similarity measurement process that compares all the models. We performed the Holm's post hoc comparison for $\alpha = 0.05$ and $\alpha = 0.10$.

## 2.8 Data transformation methods and data distribution

Data transformation attempts to change the data from one representation to another to enhance the quality thereof with a goal to enable analysis of certain information for specific purposes. In order to find out the impact of the DT methods on the datasets, data skewness and data normality measurement tests have been performed on the three different datasets and the results are visualized through Q-Q (quantile-quantile) plots [24, 27].

## 2.9 Coding and experimental environment

All experiments were conducted on a machine having Windows 10, 64-bit system with Intel Core i7 3.6GHz processor, 24GB RAM, and 500GB HD. All codes were implemented with Python 3.7. Jupyter Notebook was used for coding. All data and code are available at the following link: https://github.com/joysana1/Churn-prediction.

## 3 Results

The prediction performance of 8 classifiers combined with 6 DT methods (through rigorous experimentation on benchmark datasets) are illustrated in heatmap Fig 3. Each of these heatmaps illustrates the performance comparison (in terms of AUC, precision, recall, and F-measure) among the various CCP models for Dataset-1. Tables 10–13 in the S1 Appendix report the values for all the measures for all the datasets.

### 3.1 Results on Dataset-1

The performance of the baseline classifiers (referred to as RAW in the heatmap) on Dataset-1 is quite poor in all the metrics: the best performer in terms of F-measure is FNN with a value of 0.661 only. Interestingly, not all DT method based classifiers did perform better than RAW based classifiers. However, the performance of WOE based classifiers are consistently better than RAW classifiers in terms of both AUC and F-Measure. While RAW based FNN produces a staggering recall of 0.987, it has a poor precision (0.497), which resulted in a modest F-Measure score. The best performance is achieved by the WOE based FNN, with AUC of 0.802 and F-Measure of 0.8.

### 3.2 Results on Dataset-2

Interestingly, the performance of some baseline classifiers is quite impressive on Dataset-2, particularly in the context of AUC. For example, both DTree and GB (RAW version) achieved more than 0.82 as AUC; the F-Measure was also acceptable, particularly for GB (0.78).

Among the DT methods, again, WOE performs (in terms of F-Measure) most consistently albeit with the glitch that for DT and GB, it performs slightly worse than RAW. In fact, surprisingly enough, for GB, the best performer is RAW; for DT however, LOG and BOX-COX share the winning spot.

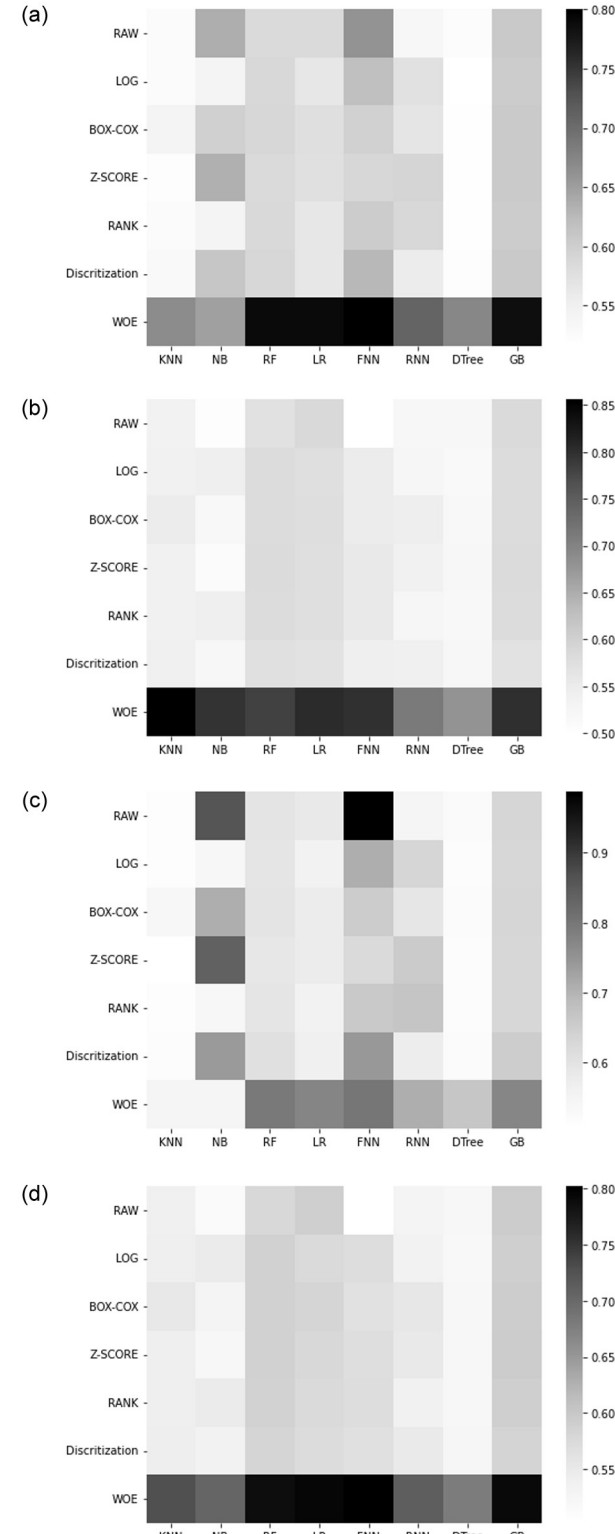

**Fig 3. Performance comparison among the CCP methods using Dataset-1. (a)** F-Measure. **(b)** Precision. **(c)** Recall. **(d)** AUC.

**Table 7. Average rankings of the algorithms.**

| Algorithm | Rank (#Position) |
|---|---|
| WOE based models | 2.4167 (#1) |
| Z-SCORE based models | 3.5417 (#2) |
| RAW based models | 3.7917 (#3) |
| Discritization based models | 4.0833 (#4) |
| BOX-COX based models | 4.1667 (#5) |
| RANK based models | 4.9375 (#6) |
| LOG based models | 5.0625 (#7) |

### 3.3 Results on Dataset-3

On Dataset-3 as well, the performance of DTree and GB in RAW mode is quite impressive: for DTree the AUC and F-Measure values are respectively 0.84 and 0.727 and for GB these are even better, 0.86 and 0.809, respectively. Again, the performance of WOE is the most consistent except in the case of DTree and GB where it is beaten by RAW. The overall winner is RF with LOG transformation which registers 0.858 for AUC and 0.82 for F-Measure.

### 3.4 Results on Dataset-4

On Dataset-4, we found very similar trend of the results—WOE consistently outperformed the RAW model as well as the models based on other DT methods. In terms of F-Measure, the best performance is achieved by FNN with WOE DT method.

### 3.5 Classifier performance analysis

Table 7 summarizes the ranking of the Freedman test among the DT methods based models across all datasets. Friedman statistic distributed according to Chi-square with ($n$-1) degrees of freedom is 24.700893. Here $n$ is the number of methods. P-value computed by the Friedman test is 0.00039. Form the Chi-square distribution table, the critical value is 12.59. Notably, 99.5% confidence interval (CI) has been considered for this test. Our Friedman test statistic value (24.700893) is greater than the critical value (12.59). So the decision is to reject the null hypothesis ($H_0$). Subsequently, the post hoc Holm test revealed significant differences among the DT methods. Fig 4 illustrates the results of Holm's test as a heat map. $p$-value $\leq 0.05$ was considered as the evidence of significance. Fig 4 tells that WOE performance is significantly different from other DT methods except for the Z-SCORE. Table 8 reflects the post hoc comparisons for $\alpha = 0.05$ and $\alpha = 0.10$. When the p-value of the test is smaller than the significant rate $\alpha = 10\%$ and 5% then Holm's procedure rejects the null hypothesis. Evidently, WOE DT based models are found to be significantly better than the other models.

## 4 Comparison with other studies

One of the datasets (Dataset-1) used in this study was also used by Amin et al. [27] and Amin et al. [42]. The study in [27] reported AUC while the authors in [42] reported F-Measure of their respective proposed models. Table 9 shows that our proposed approach performed very well compared to previously applied techniques. For this comparison, the top 2 models (in terms of F-measure as well as AUC) of this study have been selected. From the table, it is apparent that our models outperform the state-of-the-art CCP in TCI models. In particular, the improvement of the AUC metric is 26.2% in comparison to the study in [27]. In terms of F-measure, the improvement is 17% with respect to the study in [42].

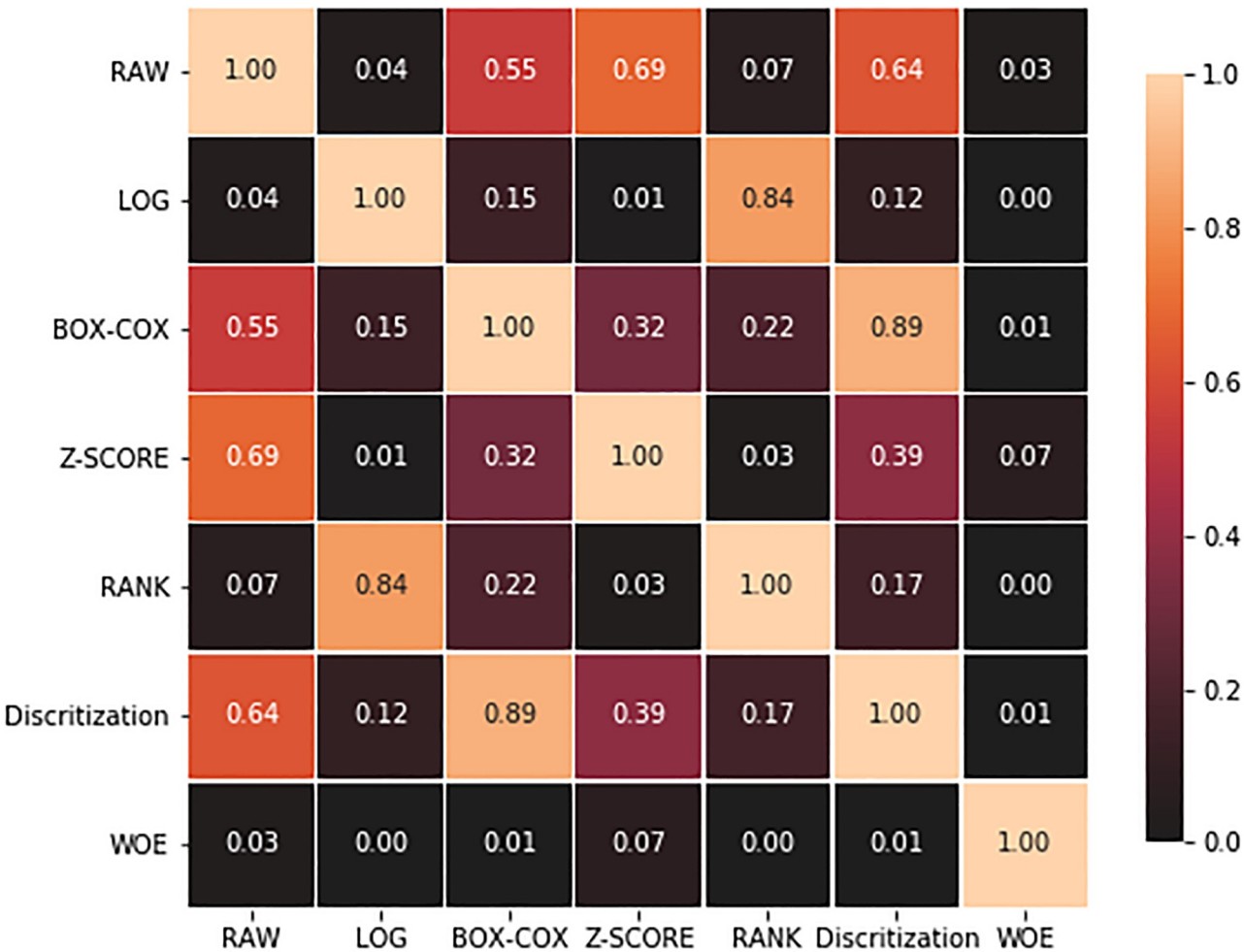

**Fig 4. Performance difference heatmap among DT based CCP models in terms of p-value.**

## 5 Impact of the data transformation methods on data distribution

The Q-Q plots are shown in Figs 5–7 for Dataset-1, Dataset-2 and Dataset-3, respectively. As we found WOE and Z-Score DT based models are performing better than the RAW (without DT) based models (see the Friedman ranked Table 7), Q-Q plots were generated only for RAW, WOE, and Z-Score methods. In each Q-Q plot, the first 3 features of the respective datasets are shown. From the Q-Q plots, it is observed that after transformation by the WOE DT method, we achieved less skewness (i.e., the data became more normally distributed).

**Table 8. Friedman and Holm test result.**

| $i$ | DT methods | p-value | Hypothesis ($\alpha = 0.05$) | Hypothesis ($\alpha = 0.10$) |
|---|---|---|---|---|
| 1 | WOE vs. LOG | 0.000022 | Rejected | Rejected |
| 2 | WOE vs. RANK | 0.000053 | Rejected | Rejected |
| 3 | WOE vs. BOX-COX | 0.005012 | Rejected | Rejected |
| 4 | WOE vs. Discretization | 0.007526 | Rejected | Rejected |
| 5 | WOE vs. RAW | 0.027461 | Rejected | Rejected |
| 6 | WOE vs. Z-SCORE | 0.071229 | Not Rejected | Rejected |

**Table 9. Performance comparison between this study and previous studies in [27] and [42] using Dataset-1.**

| Ref. Study | AUC | F-Measure |
|---|---|---|
| WOE based FNN (model in current study) | 0.802 | 0.80 |
| WOE based LR (model in current study) | 0.796 | 0.79 |
| Ref. [27] | 0.54 | - |
| Ref. [42] | - | 0.63 |

Normally distributed data is beneficial for the classifiers [27, 28]. Similar performance is also observed for Z-SCORE.

## 6 Discussion

In this paper, we have trained several classifiers to differentiate between potential churn vs. non-churn customers in the telecommunication industry (TCI). Our model building pipeline consisted of data preprocessing, data transformation, feature selection, hyperparameter tuning, model training and performance evaluation. We experimented with six different data transformation methods and eight different classification algorithms on four different datasets. From the comparative analysis and the statistical tests, it is evident that WOE transformation method in combination with state-of-the-art classification algorithms has a great impact on improving the customer churn prediction (CCP) performance in TCI. Among the classifiers investigated in this study, WOE based LR, RF, FNN, and GB classifiers showed remarkable improvement against their RAW counterparts (i.e., when no data transformation methods are applied). The data transformation techniques have shown great promise in improving the data distribution quality in general. Specially, in our experiments, the WOE method transformed data to become more normally distributed, which in the sequel provided a clear positive impact on the prediction performance for the customer churn prediction (Figs 5—7).

While a few prior works [26–28] also studied the effect of DT methods, they examined a very limited set of DT methods and experimented with only one dataset. This work, on the other hand, explores many DT methods and demonstrates the superiority of WOE across several datasets. The comparative analyses involving the RAW (without DT) based and DT based CCP models clearly suggest the potential of DT methods in improving the CCP performance (Fig 3 and Tables 10–13 in S1 Appendix). In particular, our experimental results strongly suggest that the WOE method contributed significantly towards improving the performance, except when used with DTree and GB classifiers for Dataset-2 and Dataset-3, and with GB classifier for Dataset-4. While the WOE based model failed to outperform the RAW model in these few cases, the performance of the former was quite satisfactory in any count and the degree of outperformance (by the latter) is relatively small. For example in Dataset-4, F-Measure of GB with WOE is worse than that of RAW based GB by only 4.2%. We hypothesize that this may be due to the small size of these datasets and inherent imbalance therein. The CCP prediction performance shows consistent results in favour of WOE across all classifiers on Dataset-1. This is because, for large datasets, the data variance decreases and a more reliable model may be built.

As can be seen from Table 7, WOE is the best ranked method. The post hoc comparison heatmap in Fig 4 and Table 8 reflect how the WOE performed better than the other methods. As Friedman test is rejecting the null hypothesis ($H_0$) and post hoc Holm analysis advocates the WOE based models' supremacy, it is clear that DT methods improve the user churn prediction performance significantly for the telecommunication industry.

From Table 10 in S1 Appendix, we observe that in Dataset-1, the FNN-WOE model achieves 30% and 13.9% improvement than its counterpart FNN-RAW model in terms of AUC and F-measure, respectively. Notable prediction performance improvements were also found for all

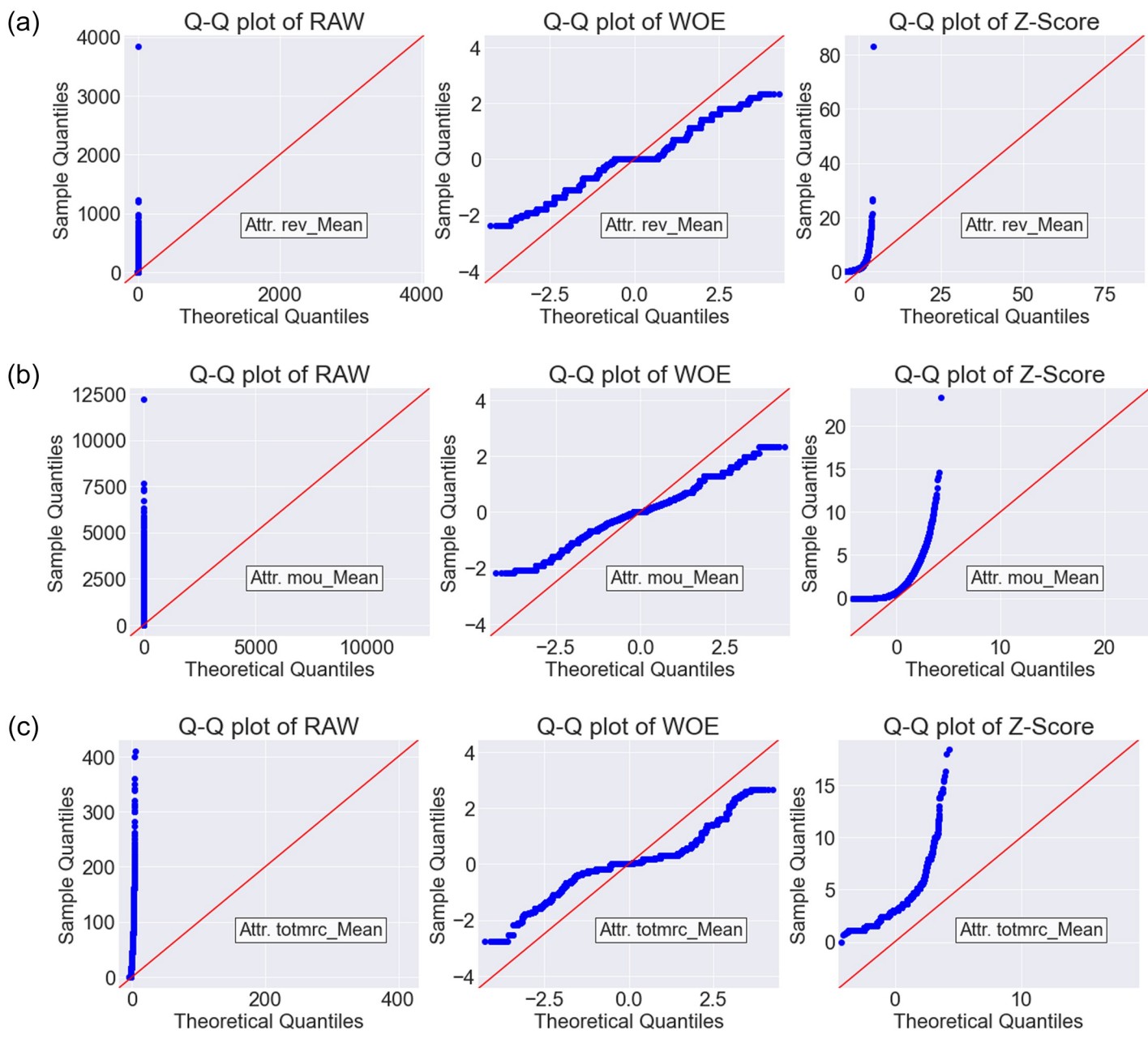

**Fig 5. The Q-Q plot for RAW (without DT), WOE and Z-Score DT methods on Dataset-1.**

the other datasets. We observe that DT methods show remarkable improvement in the performance of LR and FNN classifiers. Based on our rigorous analysis, we recommend selecting Logistic Regression (LR) or Feed-Forward Neural Networks (FNN) in association with the Weight-of-Evidence (WOE) data transformation method to construct a successful CCP model.

## 7 Conclusion

Predicting customer churn is one of the most important factors in business planning in TEL-COs. To improve the churn prediction performance, we experimented with six different data

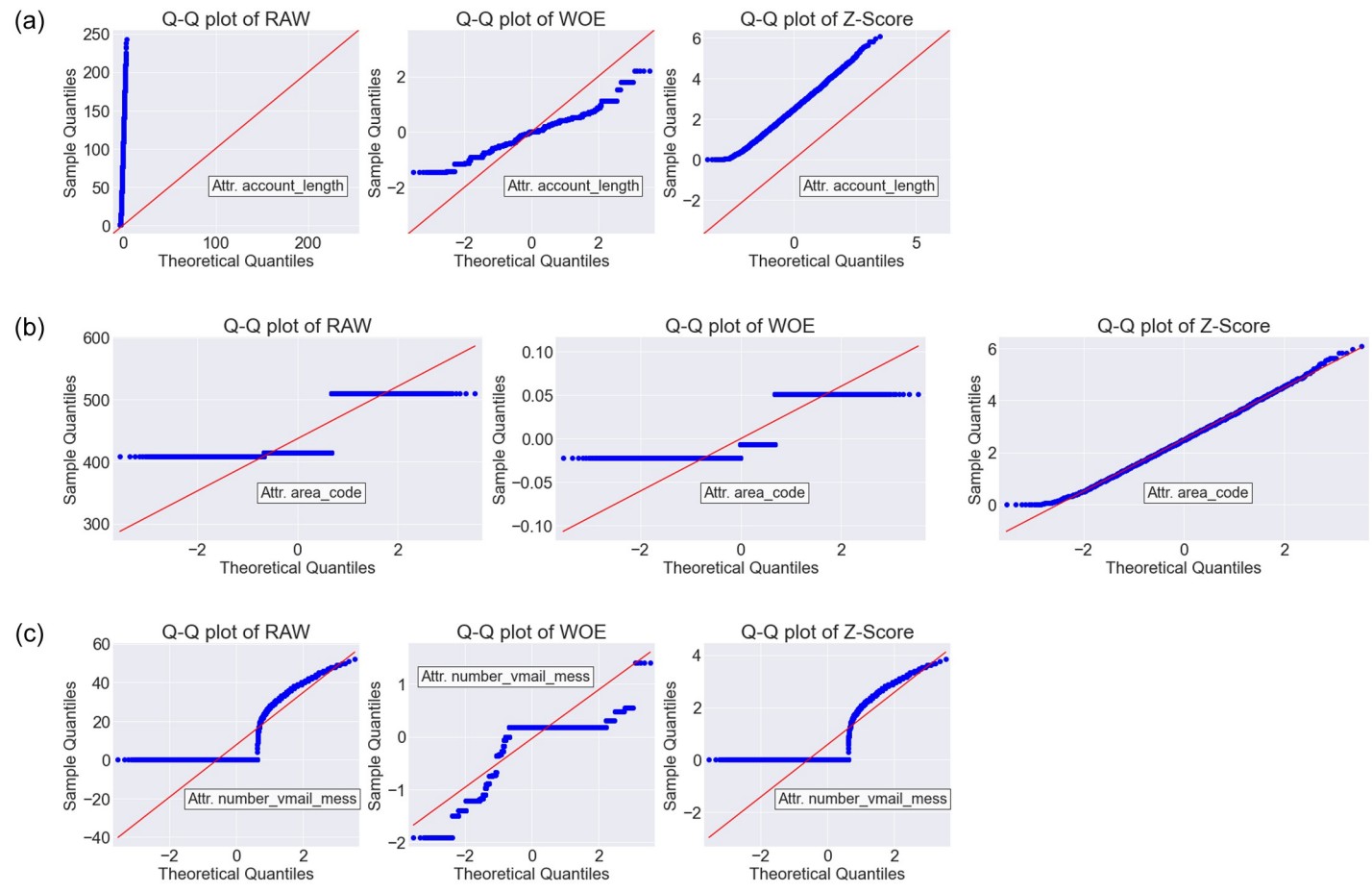

**Fig 6. The Q-Q plot forRAW (without DT), WOE and Z-Score DT methods on Dataset-2.**

transformation methods, namely, Log, Rank, Box-cox, Z-score, Discretization, and Weight-of-evidence, combined with eight different machine learning classifiers which are K-Nearest neighbor (KNN), Naïve Bayes (NB), Logistic regression (LR), Random forest (RF), Decision tree (DTree), Gradient boosting (GB), Feed-forward neural networks (FNN) and Recurrent neural networks (RNN). For each classifier, univariate feature selection method was applied to select top ranked features and grid search technique was used for hyperparameter tuning. The LR, RF, FNN, and GB are the top performing classifiers. We evaluated our methods in terms of AUC, precision, recall, and F-measure. The experimental outcomes indicate that, in most cases, the Weight-of-evidence and Z-score data transformation methods enhance the data quality and improve the prediction performance. To support our experimental results we performed Friedman non-parametric statistical test and post hoc Holm statistical analysis. The Friedman statistical test and post hoc Holm statistical analysis confirmed that Weight-of-evidence based CCP models perform better than the RAW based CCP model. Finally, we compared our proposed models with the two state-of-the-art techniques and we found that the performance of our proposed models are significantly better than that of state-of-the-art techniques. To test the robustness of our DT-augmented CCP models, we performed our experiments on both balanced (Dataset-1) and imbalanced datasets (Dataset-2, Dataset-3 and Dataset-4). In the future, we plan to extend this study with other types of data transformation

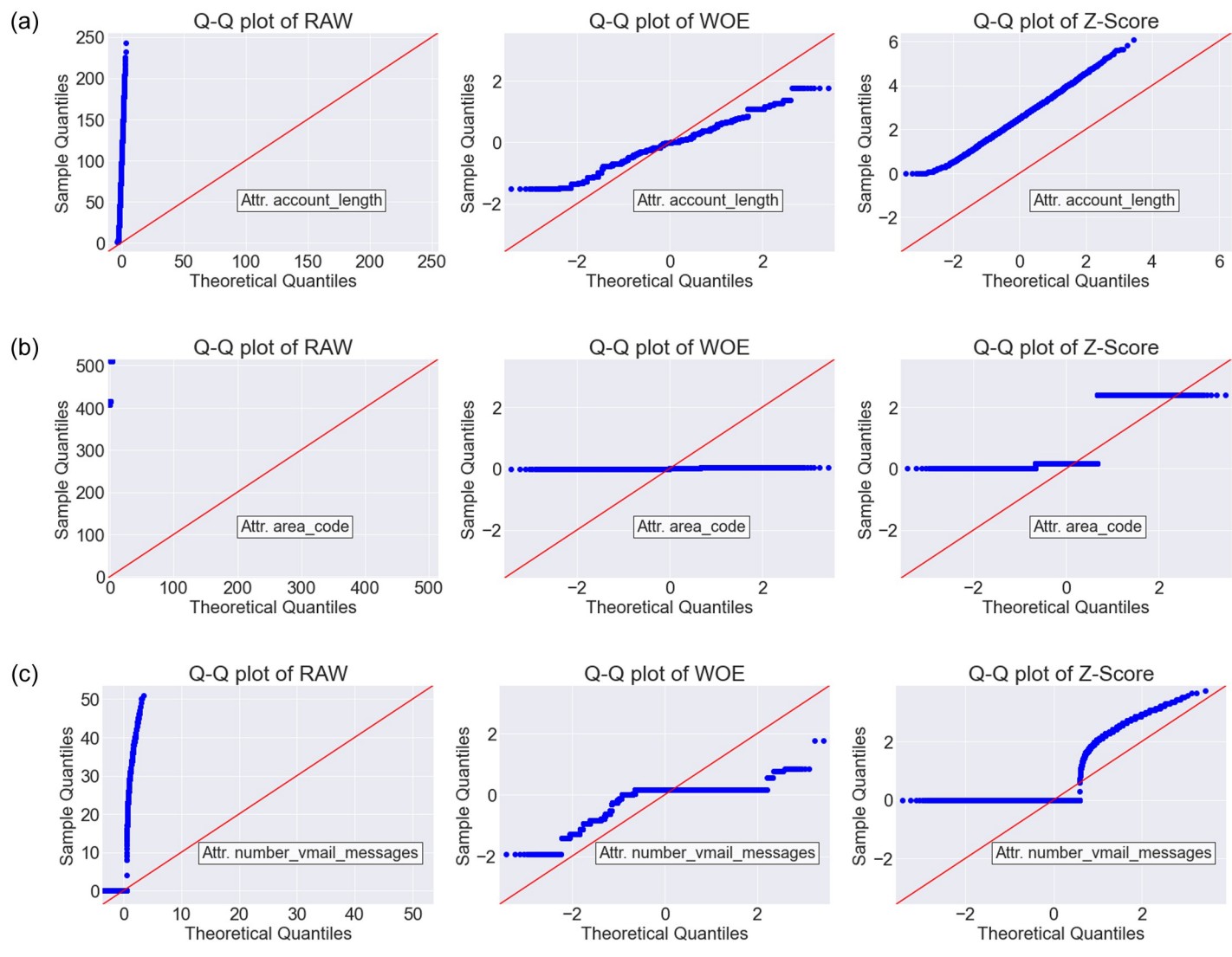

**Fig 7. The Q-Q plot for RAW (without DT), WOE and Z-Score DT methods on Dataset-3.**

approaches, classifiers, and optimization techniques. Also, our proposed learning pipeline can be tested on the other telecom datasets to examine the generalization of our results at a larger scale. Last but not the least, work can be done to extend our approach to customer churn datasets from other business sectors to study the generalization of our claim across business domains.

## Supporting information

**S1 Appendix.**
(TEX)

## Author Contributions

**Conceptualization:** Joydeb Kumar Sana, Mohammad Zoynul Abedin.

**Data curation:** Joydeb Kumar Sana.

**Formal analysis:** Joydeb Kumar Sana.

**Methodology:** Joydeb Kumar Sana.

**Project administration:** M. Sohel Rahman.

**Resources:** Mohammad Zoynul Abedin, M. Sohel Rahman.

**Software:** Mohammad Zoynul Abedin.

**Supervision:** M. Sohel Rahman, M. Saifur Rahman.

**Validation:** M. Saifur Rahman.

**Writing – original draft:** Joydeb Kumar Sana.

**Writing – review & editing:** M. Sohel Rahman, M. Saifur Rahman.

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
