## [Decision Letter · Decision Letter 0]

16 May 2022

PONE-D-22-11288A novel customer churn prediction model for the telecommunication industry using data transformation methods and feature selectionPLOS ONE

Dear Dr. Rahman,

Thank you for submitting your manuscript to PLOS ONE. After careful consideration, we feel that it has merit but does not fully meet PLOS ONE’s publication criteria as it currently stands. Therefore, we invite you to submit a revised version of the manuscript that addresses the points raised during the review process.

We look forward to receiving your revised manuscript.

Kind regards,

Ali Safaa Sadiq

Academic Editor

PLOS ONE

Journal Requirements:

Reviewers' comments:

Reviewer's Responses to Questions

**Comments to the Author**

1. Is the manuscript technically sound, and do the data support the conclusions?

Reviewer #1: Yes

Reviewer #2: Yes

Reviewer #3: Partly

Reviewer #4: Partly

2. Has the statistical analysis been performed appropriately and rigorously? 

Reviewer #1: Yes

Reviewer #2: Yes

Reviewer #3: No

Reviewer #4: I Don't Know

3. Have the authors made all data underlying the findings in their manuscript fully available?

Reviewer #1: Yes

Reviewer #2: Yes

Reviewer #3: No

Reviewer #4: Yes

4. Is the manuscript presented in an intelligible fashion and written in standard English?

Reviewer #1: Yes

Reviewer #2: Yes

Reviewer #3: Yes

Reviewer #4: Yes

5. Review Comments to the Author

Reviewer #1: The paper proposed a prediction model for the telecommunication industry using data transformation methods and feature selection; however, the article should be revised as follows:

1.English writing is good but can be improved by a native.

2.The abstract should be re-written, and principal research gaps and contributions are unclear.

3. Although the introduction is well-organised, the existing research gaps were not properly discussed and listed in the introduction section.

4. The work's main contributions and novelty can be re-written and focus mostly on the novelties.

5. In the Tables, the best-found results can be bold.

6. Please develop the section of the related works separately, and develop the current literature review using some references about the hyper-parameters tunning of deep learning model such as: a) A deep learning-based evolutionary model for short-term wind speed forecasting: A case study of the Lillgrund offshore wind farm. Energy Conversion and Management, 236, 114002. b) Short-term wind speed forecasting using recurrent neural networks with error correction, Energy, Volume 217, 2021, 119397. c) LSTM based long-term energy consumption prediction with periodicity, Energy, Volume 197, 2020. d) Prediction of electricity generation from a combined cycle power plant based on a stacking ensemble and its hyperparameter optimization with a grid-search method, Energy, Volume 227, 2021.

7. what are the benefits and drawbacks of grid search method? please add them.

8. The applied and optimised hyper-parameters should be listed in a table.

9. If it is possible provide a 3-D plot of the grid search performance for hyper-parameters tuning

Reviewer #2: 1) Customer churn prediction model for telecom using machine learning technique is not a new concept. Hence, it is not convinced that the model is novel and novelty of the model needs to be well demonstrated.

2) The organization of the manuscript should be mentioned in Introduction.

3) The literature review should emphasize both the findings and limitation. It is better to produce a comparative table.

4) The optimization of the machine learning classifiers is not well demonstrated.

5) Authors should provide more precise and critical comparison on existing related works. Need to provide more details on what is the research gap in the existing machine learning model and what are the possible ways to improve those.

6) Please revise all of the English. It is very important that the manuscript is finally revised by a native speaker.

Reviewer #3: Article Title:

A novel customer churn prediction model for the telecommunication industry using data transformation methods and feature selection

Manuscript: PONE-D-22-11288

Reviewer's Comments:

In this article, the authors have conducted a comparative study on various data transformation methods (RAW, Log, Box-cox, Rank, Discretization, Z-score, WOE) followed by feature selection (univariate feature selection, etc.). Further, they have performed hyperparameter tuning for various machine learning methods (KNN, NB, LR, RF, DTree, GB, FNN, and RNN). However, the article's contents in its current state led to an emphasis on work presentation and some technical issues listed below.

• The abstract section is very generalized and cannot reveal the clear outcomes of the proposed study.

• The literature is outdated because there is a need to cite articles from 2019 and onward.

• The proposed study may also clearly distinguish the work presented in this article from existing work https://www.sciencedirect.com/science/article/abs/pii/S0268401218305930

• The authors have used three datasets and provided the following source links:

1. https://www.kaggle.com/abhinav89/telecom-customer/data (Last Access: September 29, 2019)

2. https://data.world/earino/churn (Last Access: February 10, 2020)

3. https://www.kaggle.com/becksddf/churn-in-telecoms-dataset/data (Last Access: February 10, 2020)

However, I have observed that URL-2 and URL-3 are the same datasets. The number of samples of both datasets is different. One dataset contains 3333 samples, and the second dataset has 5000 samples. I will recommend to considered different datasets.

• I have a few observations on figure-1, which is the proposed flowchart of the optimized CCP model:

1. What is done during the preprocessing step may also be illustrated in the preprocessing block?

2. Why specifically used univariate feature selection?

3. Why straightaway terminate the process after 10-fold validation? I think 10-fold validation will produce some results which may be calculated using evaluation measures and will be used for comparison of Machine learning methods.

4. It would be more appropriate if you could add a statistical test or significance test, which is currently missing.

Reviewer #4: The contribution of this paper is good and I am happy to endorse its acceptance at some point. However, there are several major and minor comments to address. I have listed them as follows:

Please clearly state the gap targeted in this paper at the end of introduction and list down the hypotheses. In terms of research method and design, there is not much in the paper. The comparative algorithms in the experiments are not properly acknowledged and cited. I also suggest adding some figures to better articular the content as the paper looks very dry at the moment. Analysis of the results is missing in the paper. There is a big gap between the results and conclusion. There should be the result analysis between these two sections. After comparing the numerical methods, you have to be able to analyse the results and relate them to their structures. It would be interesting to have your thoughts on why the method works that way? Such analyses would be the core of your work where you prove your understanding of the reason behind the results. You can also link the findings to the hypotheses of the paper. Long story short, this paper requires a very deep analysis from different perspectives. There is no statistical test to judge about the significance of the numerical method’s results. Without such a statistical test, the conclusion cannot be supported. There is no discussion on the cost effectiveness of the proposed method. What is the computational complexity? What is the runtime? Please include such discussions. You can also use the big oh notation to show the computation complexity. Some mathematical notations and Lemma presentations are not rigorous enough to correctly understand the contents of the paper. The authors are requested to recheck all the definition of variables and further clarify these equations.

6. PLOS authors have the option to publish the peer review history of their article (what does this mean?). If published, this will include your full peer review and any attached files.

Reviewer #1: No

Reviewer #2: **Yes: **Debashish Das

Reviewer #3: **Yes: **Adnan Amin

Reviewer #4: No

---

## [Author Response · Author response to Decision Letter 0]

13 Sep 2022

Please find the response to reviewers in the ReviewResponse.pdf file.

---

## [Decision Letter · Decision Letter 1]

28 Sep 2022

PONE-D-22-11288R1A novel customer churn prediction model for the telecommunication industry using data transformation methods and feature selectionPLOS ONE

Dear Dr. Rahman,

Thank you for submitting your manuscript to PLOS ONE. After careful consideration, we feel that it has merit but does not fully meet PLOS ONE’s publication criteria as it currently stands. Therefore, we invite you to submit a revised version of the manuscript that addresses the points raised during the review process.

We look forward to receiving your revised manuscript.

Kind regards,

Ali Safaa Sadiq

Academic Editor

PLOS ONE

Journal Requirements:

Additional Editor Comments (if provided):

Authors are invited to submit their revised version of the manuscript after addressing the minor comments given by reviewer 4.

Reviewers' comments:

Reviewer's Responses to Questions

**Comments to the Author**

1. If the authors have adequately addressed your comments raised in a previous round of review and you feel that this manuscript is now acceptable for publication, you may indicate that here to bypass the “Comments to the Author” section, enter your conflict of interest statement in the “Confidential to Editor” section, and submit your "Accept" recommendation.

Reviewer #1: All comments have been addressed

Reviewer #4: (No Response)

2. Is the manuscript technically sound, and do the data support the conclusions?

Reviewer #1: Yes

Reviewer #4: (No Response)

3. Has the statistical analysis been performed appropriately and rigorously? 

Reviewer #1: Yes

Reviewer #4: (No Response)

4. Have the authors made all data underlying the findings in their manuscript fully available?

Reviewer #1: Yes

Reviewer #4: (No Response)

5. Is the manuscript presented in an intelligible fashion and written in standard English?

Reviewer #1: Yes

Reviewer #4: (No Response)

6. Review Comments to the Author

Reviewer #1: The authors have sufficiently addressed the reviewed issues in the manuscript and this work can be published.

Reviewer #4: Some final cosmetic comments:

* The results of your comparative study should be discussed in-depth and with more insightful comments on the behaviour of your algorithm on various case studies. Discussing results should not mean reading out the tables and figures once again.

* Avoid lumping references as in [x, y] and all other. Instead summarize the main contribution of each referenced paper in a separate sentence. For scientific and research papers, it is not necessary to give several references that say exactly the same. Anyway, that would be strange, since then what is innovative scientific contribution of referenced papers? For each thesis state only one reference.

* Avoid using first person.

* Avoid using abbreviations and acronyms in title, abstract, headings and highlights.

* Please avoid having heading after heading with nothing in between, either merge your headings or provide a small paragraph in between.

* The first time you use an acronym in the text, please write the full name and the acronym in parenthesis. Do not use acronyms in the title, abstract, chapter headings and highlights.

* The results should be further elaborated to show how they could be used for the real applications.

7. PLOS authors have the option to publish the peer review history of their article (what does this mean?). If published, this will include your full peer review and any attached files.

Reviewer #1: No

Reviewer #4: No

---

## [Author Response · Author response to Decision Letter 1]

31 Oct 2022

Please see attached ReviewResponse.pdf file

---

## [Decision Letter · Decision Letter 2]

10 Nov 2022

A novel customer churn prediction model for the telecommunication industry using data transformation methods and feature selection

PONE-D-22-11288R2

Dear Dr. Rahman,

We’re pleased to inform you that your manuscript has been judged scientifically suitable for publication and will be formally accepted for publication once it meets all outstanding technical requirements.

Kind regards,

Ali Safaa Sadiq

Academic Editor

PLOS ONE

Additional Editor Comments (optional):

The authors have addressed all the given comments by reviewers, hence I am happy to recommend their paper for the possible publication.

Reviewers' comments:

Reviewer's Responses to Questions

**Comments to the Author**

1. If the authors have adequately addressed your comments raised in a previous round of review and you feel that this manuscript is now acceptable for publication, you may indicate that here to bypass the “Comments to the Author” section, enter your conflict of interest statement in the “Confidential to Editor” section, and submit your "Accept" recommendation.

Reviewer #4: (No Response)

2. Is the manuscript technically sound, and do the data support the conclusions?

Reviewer #4: (No Response)

3. Has the statistical analysis been performed appropriately and rigorously? 

Reviewer #4: (No Response)

4. Have the authors made all data underlying the findings in their manuscript fully available?

Reviewer #4: (No Response)

5. Is the manuscript presented in an intelligible fashion and written in standard English?

Reviewer #4: (No Response)

6. Review Comments to the Author

Reviewer #4: all comments have been addressed. all comments have been addressed. all comments have been addressed. all comments have been addressed.

7. PLOS authors have the option to publish the peer review history of their article (what does this mean?). If published, this will include your full peer review and any attached files.

Reviewer #4: No

---

## [Editor Report · Acceptance letter]

21 Nov 2022

PONE-D-22-11288R2 

A novel customer churn prediction model for the telecommunication industry using data transformation methods and feature selection 

Dear Dr. Rahman:

I'm pleased to inform you that your manuscript has been deemed suitable for publication in PLOS ONE. Congratulations! Your manuscript is now with our production department. 

Kind regards, 

on behalf of

Dr. Ali Safaa Sadiq 

Academic Editor

PLOS ONE